# Functional Hollow Ceramic Microsphere/Flexible Polyurethane Foam Composites with a Cell Structure: Mechanical Property and Sound Absorptivity

**DOI:** 10.3390/polym14050913

**Published:** 2022-02-25

**Authors:** Jia-Horng Lin, Po-Yang Hsu, Chen-Hung Huang, Mei-Feng Lai, Bing-Chiuan Shiu, Ching-Wen Lou

**Affiliations:** 1College of Material and Chemical Engineering, Minjiang University, Fuzhou 350108, China; jhlin@fcu.edu.tw (J.-H.L.); bcshiu@mju.edu.cn (B.-C.S.); 2Laboratory of Fiber Application and Manufacturing, Department of Fiber and Composite Materials, Feng Chia University, Taichung 40724, Taiwan; 025susej@gmail.com; 3School of Chinese Medicine, China Medical University, Taichung 40402, Taiwan; 4Advanced Medical Care and Protection Technology Research Center, College of Textile and Clothing, Qingdao University, Qingdao 266071, China; 5Department of Aerospace and Systems Engineering, Feng Chia University, Taichung 40724, Taiwan; 6Department of Medical Research, China Medical University Hospital, China Medical University, Taichung 40402, Taiwan; 7Fujian Key Laboratory of Novel Functional Fibers and Materials, Minjiang University, Fuzhou 350108, China

**Keywords:** flexible polyurethane foam (FPUF), hollow ceramic microsphere (HCM), sound absorptivity, composites, functionality

## Abstract

Noise pollution is the primary environmental issue that is increasingly deteriorated with the progress of modern industry and transportation; hence, the purpose of this study is to create flexible PU foam with mechanical properties and sound absorption. In this study, hollow ceramic microsphere (HCM) is used as the filler of polyurethane (PU) foam for mechanical reinforcement. The sound absorption efficacy of PU pores and the hollow attribute of HCM contribute to a synergistic sound absorption effect. HCM-filled PU foam is evaluated in terms of surface characteristic, mechanical properties, and sound absorption as related to the HCM content, determining the optimal functional flexible PU foam. The test results indicate that the presence of HCM strengthens the stability of the cell structure significantly. In addition, the synergistic effect can be proven by a 2.24 times greater mechanical strength and better sound absorption. Specifically, with more HCM, the flexible PU foam exhibits significantly improved sound absorption in high frequencies, suggesting that this study successfully generates functional PU foam with high mechanical properties and high sound absorption.

## 1. Introduction

Noise pollution is of the major pollutions in the environment and increasingly escalating with the progress of industries and transportation. Noise is commonly divided into structure-borne noise caused by engines or other objects when in contact with the ground while air-borne noise caused by high-speed wind [1]. The main mechanism to reduce noise pollution involves a rise in abrasion against air molecules, as well as the damping of absorbent materials. Therefore, the sound absorption frequency is the main concern for the design of sound absorbing materials. Playing a crucial role, the damping can absorb the sound or delay the transmission of sound, during which the acoustic energy is transformed into thermal energy, thereby achieving effective sound absorption and noise abatement [2]. 

Polyurethane (PU) elastomer has been the most frequently used industrial polymer [3], so PU made of isocynate and polyol is the most popular material used in the construction, automobile, sports, navy, and furniture fields [4,5,6,7]. PU foam can be classified into rigid polyurethane foam (RPUF) and flexible polyurethane foam (FPUF). Composed of more polyol than RPUF, FPUF demonstrates greater flexibility, which means that FPUF outperforms RPUF in withstanding a high deformation. In addition, FPUF also possesses excellent physical properties and cushioning effect, so it is pervasively used in different regions; e.g., furniture, carpet, and vehicles [8,9]. 

There are a great number of studies on the modification of PU foam, including two groups. One group focuses on the change of the inner chemical structure. For example, Sung et al. confirmed that the sound absorptivity of materials was improved when the foaming structure was changed [2]. Moreover, Tian et al. proved that the incorporation of polyimide had a positive influence over the stability of the foaming materials [10]. The other group emphasizes the incorporation of filler with foaming materials. Estravis et al. indicated that the use of particulate material as filler provided PU foam polymer with a basic function [4]. Moreover, Sung et al. reported that when used as a filler, hydroxide could change the cell structure, strengthening the sound absorptivity of materials at different frequencies [11]. There have been scholars using different fillers to change the properties of the composites in recent years [12,13,14,15,16,17,18]. To summarize, either a change in chemical structure of materials or a reinforcement of fillers can provide PU foam with a synergistic effect, and the methods previously described are proved to be effective in attaining the functions as expected. 

Hollow ceramic microsphere (HCM) is a filler with a light weight [19], a low density, a bioinert attribute, a good thermal conductivity, a great distribution, and an excellent acoustic effect. Subsequently, HCM is a popular item used in the construction, aviation, chemistry, and electronics industries [20,21,22,23,24], but there are few studies focusing on using HCM in polyurethane foam. In this study, HCM is thus used as the filler for flexible polyurethane foam (FPUF), thereby joining the hollow attribute with the cell-filled PU foam while achieving a synergistic effect in sound absorptivity and mechanical reinforcement. Finally, the surface characterization, mechanical property, and sound absorptivity of the PU foam composites are evaluated as related to the content of HCM, determining the optimal functional HCMF sound absorbent composites.

## 2. Materials and Methods

### 2.1. Materials

High density soft PU is made of agent A (polyol) and B (isocynate) that were purchased from Kuang Lung Shing Co., Taipei, Taiwan. Hollow ceramic microsphere (HCM) was purchased from Feng Chia University, Taiwan.

### 2.2. Preparation

In this study, the hollow ceramic microsphere (HCM) was used as a filler to produce the sound absorbent soft PU foam. HCM (0, 5, 10, 15, and 20 wt%) was separately added to the agent A for a 10-min pre-mix, after which PU curing agent was added for another 10-s mixing. The blends were infused into a mold for the 3-h foaming at normal temperature. After curing, the PU foam was demolded to yield HCM-contained flexible PU foam (FPUF) composites. Next, the surface observation, mechanical properties, and sound absorption measurements were conducted in order to obtain the optimal HCM/FPUF sound absorbent composites. Figure 1 and Table 1 shows the preparation method and sample code, respectivey.

### 2.3. Characterization

#### 2.3.1. Surface Observation

A stereomicroscope (SZ-CTV, YUANYU Group CO., Ltd., Taipei Taiwan) and a field emission scanning election microscope (SEM, S-4800, HITACHI, Tokyo, Japan) were used for the morphology observation. 

The stereomicroscope was used to observe the macro cell structure of samples that are dyed in advance. The difference in the cell diameter was compared using Image Pro Plus that was provided by Feng Chia University, Taiwan, and analyzed accordingly. By contrast, the SEM was used to compare the micro-structure of the sound absorbent composites, as well as the damage rate of the cell structure as related to the content of HCM. Magnification for the stereomicroscope was 10 × 0.67 and 10 × 4.00, while the magnification for SEM was 8.00 mm × 50 and 8.00 mm × 500.

#### 2.3.2. Compressibility Test

As specified in ASTM D1621-10 (Standard Test Method for Compressive Properties of Rigid Cellular Plastics), HCMF composites were compressed to 25% of the thickness at a test rate being 1 mm/min, thereby examining the compression strength. Six samples for each specification were used, and the sample size was 50 mm × 50 mm × 20 mm.

#### 2.3.3. Drop Weight Impact Test

As specified in ASTM D4168-95 (2008) E1 (Standard Test Methods for Transmitted Shock Characteristics of Foam-in-Place Cushioning Material, HCMF composites were tested for the impact effect. Samples were trimmed into pieces of 10 cm × 10 cm, after which the impactor was released from a specified height of 307 cm to impact the sample. The impact force and impact resistance were then recorded to evaluate the impact performance of samples.

#### 2.3.4. Sound Absorptivity Test

As specified in ASTM E1050-12 (Standard Test Method for Impedance and Absorption of Acoustical Materials using a Tube, Two Microphones and a Digital Frequency Analysis System) as Figure 2, the sound absorption coefficient of HCMF composites in a frequency range of 125–4000 Hz was evaluated. The temperature was 24 °C while the relative humidity was 50%. Samples were trimmed into cylindrical pieces with a diameter of 38 mm, and three samples for each specification were used. A sample was inserted into the impedance tube for the first measurement, after which the microphones were switched in position for the second measurement, thereby rectifying the difference between two microphones. The sound absorption coefficient-frequency curves were plotted automatically afterwards.

## 3. Results and Discussion

### 3.1. Stereomicroscopic Observation

Figure 3 shows the stereomicroscopic images where HCMF composites exhibit a cell density that becomes compact with a rise in the HCM content, the results of which are in conformity with the SEM observation in Figure 4. The SEM images clearly demonstrate that the cell size distinctively appears consistent with the increasingly greater HCM content. Hence, it is substantiated that the presence of HCM is correlated with the cell structure and morphology, stabilizing the cell morphology [25,26]. In addition, sound absorptivity of composites is dependent on the difference in the cell structure.

### 3.2. Effects of HCM on Cell Size of Flexible PU Foam

Figure 5 shows that with 5wt% of HCM, the PU foam exhibits a more unstable cell size distribution than the groups with 10–20 wt% of HCM. The 5wt% group has a cell size that is distributed between 300 µm and 1000 µm. The greater the HCM content, the denser the cell diameter distribution. When there is a higher HCM content, there is a greater amount of HCM embedded in the cell walls, so HCM are distributed more evenly. A low content of HCM hampers an even HCM distribution over the cell walls, which means that the cell size exhibits instability; namely, cell size has a great range. This phenomenon subsequently affects the mechanical property and sound absorptivity [16,27,28]. Furthermore, Figure 6 also indicates that when the HCM content exceeds 5wt%, the cell size of the flexible PU foam (FPUF) is also decreased, which supports the description for Figure 5. In general, the cell size of FPUF is correlated with the HCM content, as exemplified by the decreasing error range in Figure 6. To summarize, a low HCM content adversely affects cell growth, resulting in a greater range of cell size, while a high HCM content helps stabilize the cell size, strengthening the density of FPUF.

### 3.3. SEM Observation

According to Figure 7A–E, the cell size of composites has a decreasing trend with a rise in the HCM content. The SEM images indicate that the composites acquire better cell integrity with the presence of more HCM. When the HCM content is increased, HCM can be better distributed over FPUF, improving the quality of the cell walls. During the foaming reaction, a better quality of cell walls generates a greater number of smaller cells, instead of the occurrence of over-foaming. In other words, cell walls with better quality help improve the cell density. Figure 7F–J shows that a greater HCM content equivalently means a better distribution of HCM, which is consistent with the findings in Figure 5 and Figure 6.

### 3.4. Effect of HCM Content over Compressibility of Composites

Figure 8 shows the compression strength of HCMF composites that is in direct proportion to the HCM content. Specifically, with 20wt% of HCM, the composites exhibit 2.24-times greater compression strength than the control group, which suggests that the presence of filler has a significantly positive influence over the compression strength of HCMF composites. HCM is inorganic ceramic that absorbs and disperses much of a compression force. When FPUF is incorporated with HCM, the latter of plays a role of hard chain segments when being embedded in the cell walls of PU foam during the foaming and curing process [16,17,25]. A high HCM content means a greater amount of HCM. In particular, 20wt% of HCM provides the composites with a maximal compression strength as HCM reaches the critical value of withstanding a compression force. The distinct improvement in compression strength can apply a comparable fluid mechanism of shear thickening fluid (STF). When STF encounters a stress, the fluid particles agglomerate to generate a thickening phenomenon that can bear a higher stress (Figure 9) [29].

These conclusions were confirmed with the Mohammed Imran work. Notably, scholars have yet to use HCM as form fillers to improve the material property (detailed information can be found in introduction). As a result, the suggested relationship between HCM and mechanical performance could be one directive principle for designing desired durable form materials [17].

Before being exerted as a force, the microspheres of STF are dispersed as Figure 9a, and then in response to an external force, STF demonstrates agglomeration that resists an impact force or a stab force, as in Figure 9b. Although HCM in this study is unlike STF that reacts with agglomeration against an impact force, a high content of HCM still causes a higher filler density that achieves even distribution over cell walls. In the meanwhile, a high HCM content strengthens the density of composites, which contributes to an agglomeration-like effect that strengthens the compression resistance. Comparing HCMF0 and HCMF20 in terms of being exerted by an external force, as in Figure 10a,b, the filler (i.e., HCM) is stacked as STF when being compressed, thereby distributing the compression force that HCM bears. Hence, the compression strength of composites is improved when there is a greater content of HCM.

### 3.5. Effect of HCM Content over Impact Resistance of Composites

Figure 11 shows the impact strength of HCMF composites, which is 2579.96 N for HCMF0, 3018.03 N for HCMF5, 2760.45 N for HCMF10, 1969.47 for HCMF15, and 1932.64 N for HCMF20, respectively. As the impact force is exerted over the composites vertically, the lower the impact force, the better the impact strength. When the content of HCM is higher, the proposed materials demonstrate a lower impact strength—namely, a better impact resistance. The presence of HCM is substantiated to have a positive influence over the impact resistance of composites. As described in Figure 10, HCM that is distributed over the cell walls is able to disperse an impact force, which indicates that HCMF composites exhibit a significantly lower impact strength when composed of a greater amount of HCM. Sujon et al. reviewed and commented that the incorporation of nanoparticles improved the volume of interfacial area, which subsequently improved the energy dissipation. The deformation, relaxing, and regeneration molecular chain networks cause the viscoelasticity damping, which suggests a significant correlation between frequency and molecular motion. Hence, the molecular structure of polymer bonds particles fulfill the function of elastic material that constrain particles inside the material from bouncing [30]. This finding further proves that the incorporation of HCM as the filler provides damping of structures.

On the other hand, the incorporation of filler is also pertinent to the viscoelasticity of PU foam [26], so the viscoelasticity of cell walls is dependent on the presence of HCM. Furthermore, a rise in the content of filler may lead to agglomeration that interferes with the impact resistance of the composites as previously described.

### 3.6. Effect of HCM on Sound Absorption Coefficient of Composites

Figure 12 shows that HCMF composites demonstrate an unstable sound absorption coefficient when the sound waves are at frequencies of 125–1000 Hz. Consisting of a lower content of HCM, HCMF5 and HCMF10 acquire a sound absorption coefficient that is comparatively higher than that of HCMF15 and HCMF20 that are composed of a greater content of HCM, as well as the control group of HCMF0. When the frequency exceeds 1500 Hz, sound absorbent composites show an increasing trend in the sound absorption coefficient with the content of HCM being increased. When at 2000 Hz, HCMF20 has a higher sound absorption coefficient than HCMF0, and when at 3500 Hz, HCMF15 also has a higher sound absorption coefficient than HCMF0. To summarize, it is a small amount of HCM that affects the sound absorptivity of sound absorbent composites in the range of 125–1500 Hz, while it is a great amount of HCM that affects the sound absorptivity of sound absorbent composites in the range of 2000–4000 Hz.

The major factor of sound absorption coefficient is a steady increase in the HCM content that improves the cell walls of PU foam. It is part of the process that when HCM is embedded into the cell walls to eventually fill the cells of PU foam, the sound absorptivity of composites changes accordingly. The average sound absorptivity is comparatively lower when composites are composed of a small amount of HCM. This result may be ascribed to a sporadic distribution of HCM in the PU foam, which in turn causes a greater range of cell size, as shown in Figure 3. This specific group also has a comparatively wider cell size distribution than the control group, as well as the other groups containing a greater amount of HCM.

When the filler content increased, this study became similar to the previous work (Baek, Seung Hwan [25]) in sound absorption property. The sound absorption frequency increases in medium high frequency with further increments of filler content.

Since the HCM content increases by degrees, PU foam starts to obtain a higher cell density and a more stabilized cell structure, which benefits the corresponding sound absorptivity. Also, the HCMF sound absorbent composites that are composed of a greater amount of HCM are proved to exhibit higher sound absorption coefficients at high frequencies. Therefore, the incorporation of HCM as the filler for PU foam is effective in improving the sound absorptivity of composites in this study, exemplifying a synergistic mechanism [13,14,15,18,25,31]. Similarly, Sung et al. studied the difference in the sound absorptivity of PU foam, which was found to be dependent on the addition of hollow ceramic microspheres, as well as the resulting structural resonance. The findings also confirmed the difference in sound absorptivity that was caused by the different damping with corresponding structure [11].

## 4. Conclusions

In order to obtain functional hollow ceramic microsphere/flexible polyurethane foam composites, this study examines the proposed HCMF sound absorbent composites in terms of surface observation, cell size analysis, compression force, and sound absorptivity. As for the morphology observation, the presence of HCM changes the cell structure of composites, and the cell size shows a significant decreasing trend when the content of HCM is increased. According to the SEM observation, the higher the HCM content, the greater the HCM amount embedded in the cell walls. As for the maximal compressibility of composites, it reaches 317.62 N, which is 2.24 times higher than that of HCMF0, while the sound absorption coefficient of composites is higher than 0.45 at high frequencies. Moreover, HCMF5 demonstrates an optimal impact resistance as high as 3018.03 N. To summarize, the incorporation of HCM helps improve the cell structure, mechanical properties, and functions of HCMF sound absorbent composites, which suggests that this study has successfully produced the functional composites made of a hollow ceramic microsphere and flexible polyurethane foam.

## Figures and Tables

**Figure 1 polymers-14-00913-f001:**
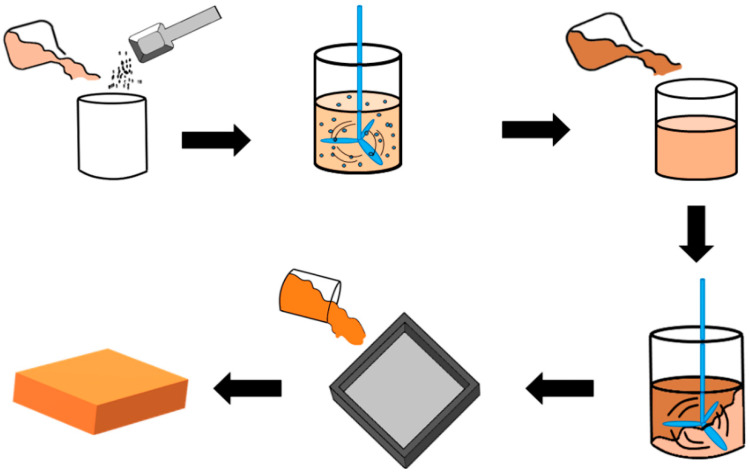
Illustrative diagram of the preparation process HCMF composites.

**Figure 2 polymers-14-00913-f002:**
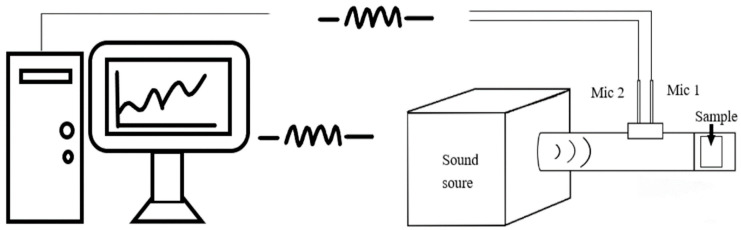
Diagram of two microphones method for sound absorption coefficient measurement.

**Figure 3 polymers-14-00913-f003:**
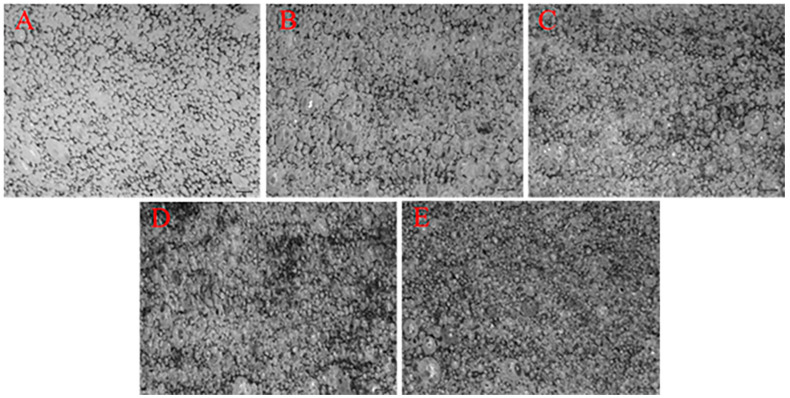
Stereomicroscopic images (magnified by 10 × 0.67) of (**A**) HCMF0, (**B**) HCMF5, (**C**) HCMF10, (**D**) HCMF15, and (**E**) HCMF20.

**Figure 4 polymers-14-00913-f004:**
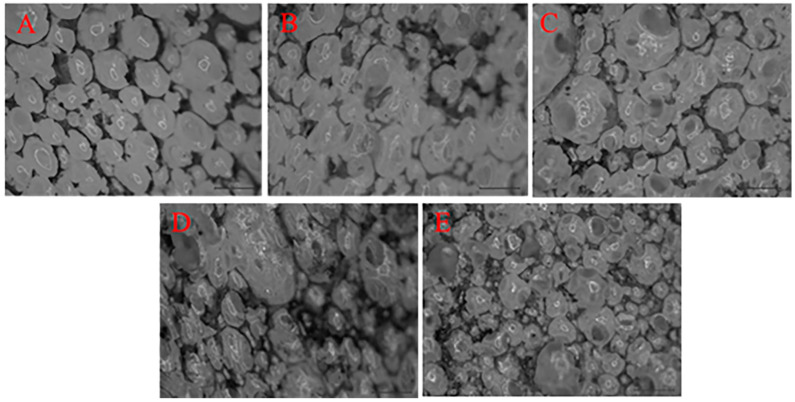
Stereomicroscopic images (magnified by 10 × 4) of (**A**) HCMF0, (**B**) HCMF5, (**C**) HCMF10, (**D**) HCMF15, and (**E**) HCMF20.

**Figure 5 polymers-14-00913-f005:**
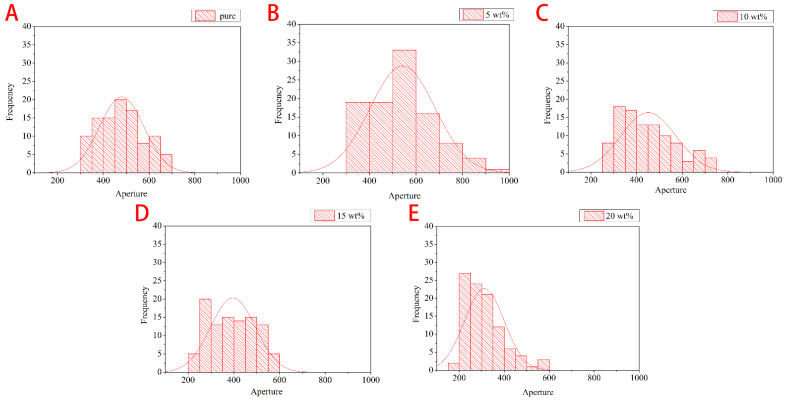
The aperture distribution (µm) of (**A**) HCMF0, (**B**) HCMF5, (**C**) HCMF10, (**D**) HCMF15, and (**E**) HCMF20 as related to the HCM content.

**Figure 6 polymers-14-00913-f006:**
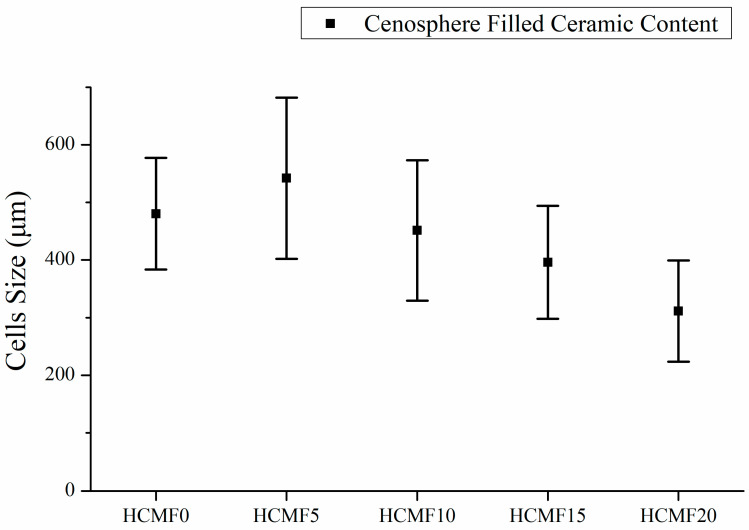
Average cell size of HCMF composites as related to the HCM content.

**Figure 7 polymers-14-00913-f007:**
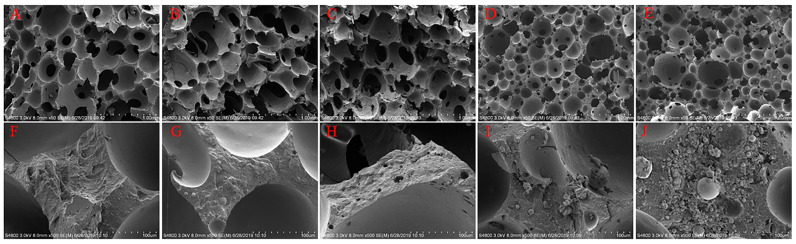
SEM images of the control group (**A**–**E**) and the experimental group (**F**–**J**) containing 5, 10, 15, and 20 wt% of HCM. Row 1 and 2 separately has a magnification of 8.00 mm × 50 and 8.00 mm × 500.

**Figure 8 polymers-14-00913-f008:**
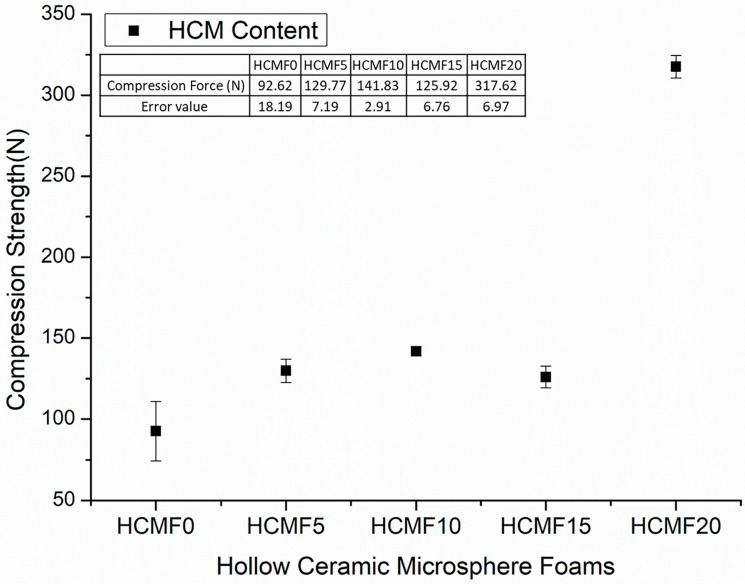
Compression force of HCMF composites as related to the HCM content.

**Figure 9 polymers-14-00913-f009:**
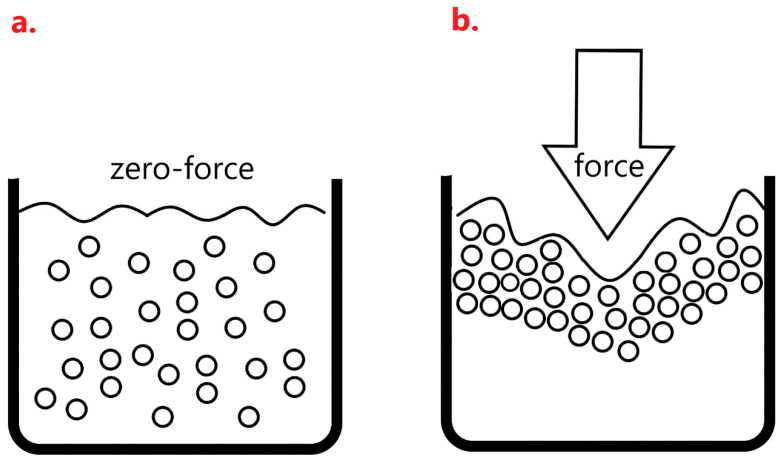
The difference in STF as to whether a compression force is exerted. (**a**) zero-force; (**b**) Force.

**Figure 10 polymers-14-00913-f010:**
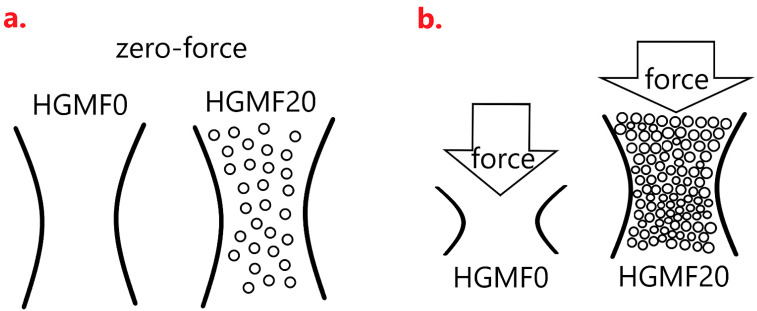
The difference in cell wall whether a compression force is exerted. (**a**,**b**) shows the schematic diagram of samples before and after stress force loading.

**Figure 11 polymers-14-00913-f011:**
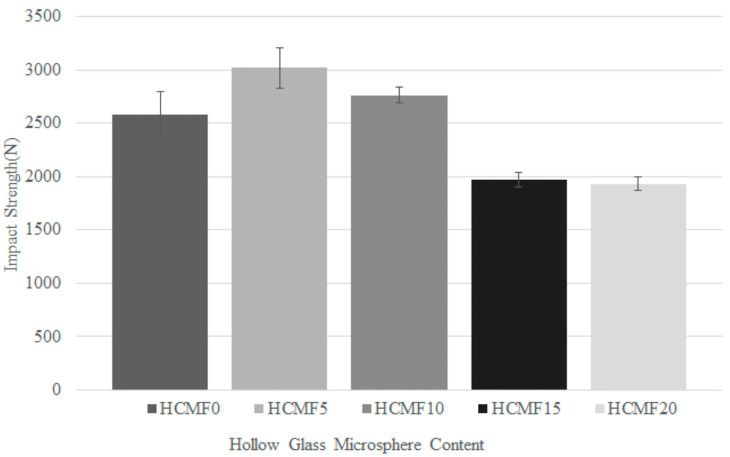
Impact resistance of HCMF composites as related to the HCM content.

**Figure 12 polymers-14-00913-f012:**
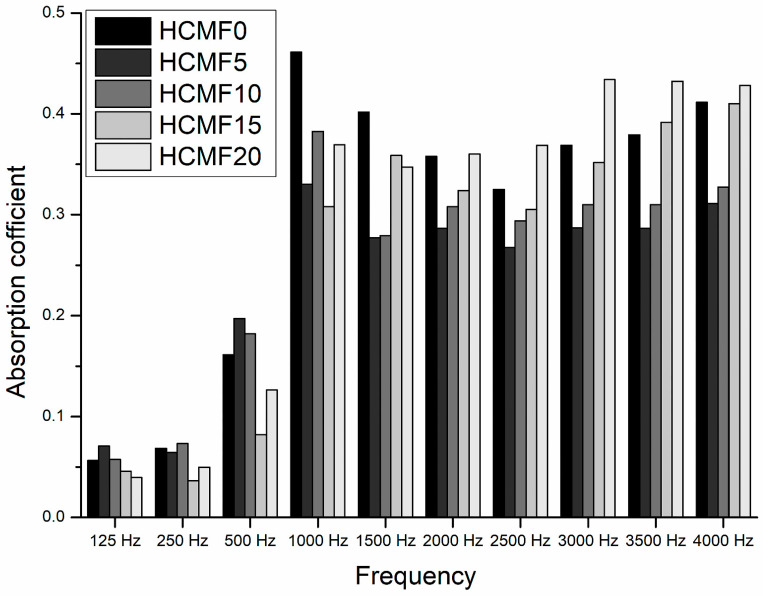
Sound absorption coefficient of HCMF composites as related to the HCM content.

**Table 1 polymers-14-00913-t001:** Denotation and specification of HCMF composites.

	HCMF0	HCMF5	HCMF10	HCMF15	HCMF20
PU-A	80	77.5	75	72.5	70
PU-B	20	17.5	15	12.5	10
HCM	0	5	10	15	20

## Data Availability

All data relevant to the study are included in the article.

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
