# Peer review of "Functional Hollow Ceramic Microsphere/Flexible Polyurethane Foam Composites with a Cell Structure: Mechanical Property and Sound Absorptivity"

_polymers, 2022, doi:10.3390/polym14050913_

Round 1

Reviewer 1 Report

The manuscript under the title: “Functional Hollow Ceramic Microsphere/Flexible Polyurethane Foam Composites with a Cell Structure: Mechanical Property and Sound Absorptivity” is in line with the Polymer journal. The article was based on original. The article is overall quite well described, but requires major improvements before being considered for publication, especially:

  • Abstract: please define main aim of the research.
  • Abstract: lack of information about the results obtained and the most important findings.
  • Introduction: please define the gap in the literature that justified selected topic.
  • Introduction: please stress the novelty of provided research.
  • Chapter 2.2.: please add more detailed description of the process of the samples’ temperatures, including time and temperatures.
  • Chapter 2.3.1.: Add information about the magnifications used.
  • Chapter 3: please do not start chapter from figures.
  • Figure 8: the unit should be MPa or kPa.
  • Chapter 3.4. What was the criterium for the end of the test. Could you also enclosed some pictures after the tests and describe their behaviour?
  • Discussion: Lack of discussion with the up-to-date literature, compare obtained results with other research works, add new literature.
  • Conclusions; please define the most important measurable results.

Author Response

Manuscript ID polymers-1573690

Functional Hollow Ceramic Microsphere/Flexible Polyurethane Foam Composites with a Cell Structure: Mechanical Property and Sound Absorptivity

Polymers

Reviewer 1

Comments and Suggestions for Authors

The manuscript under the title: “Functional Hollow Ceramic Microsphere/Flexible Polyurethane Foam Composites with a Cell Structure: Mechanical Property and Sound Absorptivity” is in line with the Polymer journal. The article was based on original. The article is overall quite well described, but requires major improvements before being considered for publication, especially:

  1. Abstract: please define main aim of the research.
  2. Abstract: lack of information about the results obtained and the most important findings.

Answer: Thank you for your comment. The answer is given according to the previous two questions. The abstract has now been revised as follows.

The noise pollution is the primary environmental issue that is increasingly deteriorated with the progress of modern industry and transportation, hence the purpose of this study is to create flexible PU foam with mechanical properties and sound absorption. In this study, hollow ceramic microsphere (HCM) is used as the filler of polyurethane (PU) foam for mechanical reinforcement. The sound absorption efficacy of PU pores and the hollow attribute of HCM contribute to a synergistic sound absorption effect. The HCM-filled PU foam are evaluated in terms of surface characteristic, mechanical properties, sound absorption as related to the HCM content, determining the optimal functional flexible PU foam.

The test results indicate that the presence of HCM strengthens the stability of the cell structure significantly. In addition, the synergistic effect can be proven by a 2.24 times greater mechanical strength and a better sound absorption. Specifically, with more HCM, the flexible PU foam exhibits significantly improved sound absorption in high frequencies, suggesting that this study successfully generates functional PU foam with high mechanical properties and high sound absorption.

 Introduction: please define the gap in the literature that justified selected topic.

Answer: Thank you for your comment. Line 54-67 describes the excellent contribution of fillers to samples while line 68-72 stresses that there are few studies using HCM as filler, thereby clarifies the difference between previous studies and this study.

  1. Introduction: please stress the novelty of provided research.

Answer: Thank you for your comment. The novelty of using HCM to the PU foam field has been stated in line 68-72, which provides this study with an innovative perspective.

  1. Chapter 2.2.: please add more detailed description of the process of the samples’ temperatures, including time and temperatures.

Answer: Thank you for your comment. The supplementations and revisions are marked in green.

In this study, hollow ceramic microsphere (HCM) is used as filler to produce the sound absorbent soft PU foam. HCM (0, 5, 10, 15, and 20 wt%) is separately added to the agent A for a 10-minute pre-mix, after which PU curing agent is added for another 10-second mixing. The blends are infused into a mold for the 3-hour foaming at normal temperature. After curing, the PU foam is demolded to yield HCM-contained flexible PU foam (FPUF) composites. Next, the surface observation, mechanical properties, and sound absorption measurements are conducted in order to obtain the optimal HCM/FPUF sound absorbent composites.

  1. Chapter 2.3.1.: Add information about the magnifications used.

Answer: Thank you for your comment. The magnification for stereomicroscope is 10*0.67 and 10*4.00 while the magnification for SEM is 8.00mm*50 and 8.00mm*500.

  1. Chapter 3: please do not start chapter from figures.

Answer: Thank you for your comment. This format mistake has now been modified.

  1. Figure 8: the unit should be MPa or kPa.

Answer: Thank you for your comment. The authentic unit is N in this study.

  1. Chapter 3.4. What was the criterium for the end of the test. Could you also enclosed some pictures after the tests and describe their behaviour?

Answer: Thank you for your comment. The test standard used in section 2.3.2 is ASTM D1621-10, and the behavior of samples when a force is exerted is demonstrated in Figures 9 and 10. As the actual internal deformation cannot be observed, it is simulated in the two figures.

  1. Discussion: Lack of discussion with the up-to-date literature, compare obtained results with other research works, add new literature.

Answer: Thank you for your comment. More discussion and information have now been appended to section 3.5. Sujon et al. reviewed and commented that the incorporation of nanoparticles improved the volume of interfacial area, which subsequently improved the energy dissipation. The deformation, relaxing, and regeneration molecular chain networks cause the viscoelasticity damping, which suggests the significant correlation between frequency and molecular motion. Hence, the molecular structure of polymer bonds particles, fulfilling the function of elastic material that constrain particles inside the material from bouncing. This finding further proves that the incorporation of HCM as the filler provides damping of  structures in fact.

  1. Conclusions; please define the most important measurable results.

Answer: Thank you for your comment. More information regarding measurements has now been appended to the text as follows.

The maximal compressibility of composites reaches 317.62 N that is 2.24 times higher

than that of HCMF0; the sound absorption coefficient of composites is higher than

0.45 at high frequencies. Moreover, HCMF5 demonstrates an optimal impact resistance as high as 3018.03 N

Reviewer 2 Report

Interesting work, well-presented and clearly exposed.

I raise some minor remarks:

line 25, please specify what HCM stands for

Avoid 'flexible' as Keyword

Lines 36-38: please rewrite, two types of noise are not clearly evidenced

Line 59 and 62: avoid reporting the year of publication after the reference's citation

Line 99-100: please revise english

Line 212-219 font is different

Line 222: Capital letter in caption is missing

Author Response

Manuscript ID polymers-1573690

Functional Hollow Ceramic Microsphere/Flexible Polyurethane Foam Composites with a Cell Structure: Mechanical Property and Sound Absorptivity

Polymers

Reviewer 2

Comments and Suggestions for Authors

Interesting work, well-presented and clearly exposed.

I raise some minor remarks:

  1. line 25, please specify what HCM stands for

Answer: Thank you for your comment. HCM refers to hollow ceramic microspheres that have now been appended to the text.

  1. Avoid 'flexible' as Keyword

Answer: Sorry for the oversight. The keyword has now been revised as “flexible polyurethane foam (FPUF).”

More information regarding measurements has now been appended to the text as follows.

Keywords: flexible polyurethane foam (FPUF); hollow ceramic microsphere (HCM); sound absorptivity; composites; functionality

  1. Lines 36-38: please rewrite, two types of noise are not clearly evidenced

Answer: Thank you for your comment. The supportive literature has now been added as follows.

Gwon, J.G., et al., Sound absorption behavior of flexible polyurethane foams with distinct cellular structures. 2016. 89: p. 448-454.

  1. Line 59 and 62: avoid reporting the year of publication after the reference's citation

Answer: Thank you for your comment. The revisions have now been implemented.

  1. Line 99-100: please revise English

Answer: Thank you for your comment. The correct content has now been provided.

The difference in the cell diameter is compared using the Image Pro Plus that is provided by Feng Chia University, Taiwan, and analyzed accordingly.

  1. Line 212-219 font is different

Answer: This mistake has now been modified as required.

  1. Line 222: Capital letter in caption is missing

Answer: Thank you for your comment. This error has been revised.

Round 2

Reviewer 1 Report

  • The revised manuscript - entitled: “Functional Hollow Ceramic Microsphere/Flexible Polyurethane Foam Composites with a Cell Structure: Mechanical Property and Sound Absorptivity” were improved by the authors. However the some comments still requires changes in the article:

    • Introduction: The gap should be clearly defined in the last paragraph of the Introduction part and supported by proper literature. In this moment, used literature is not sufficient and the novelty aspects are not well stressed.  
    • Discussion: The discussion is still very generic. This topic required wider  discussion with the up-to-date literature, compare obtained results with other research works, add new literature.

Author Response

Manuscript ID polymers-1573690

Functional Hollow Ceramic Microsphere/Flexible Polyurethane Foam Composites with a Cell Structure: Mechanical Property and Sound Absorptivity

Polymers

Reviewer

  1. Introduction: The gap should be clearly defined in the last paragraph of the Introduction part and supported by proper literature. In this moment, used literature is not sufficient and the novelty aspects are not well stressed.

Answer: Thank you for your comment. Regarding the description of “Wang et al. indicated that the presence of expanded graphene swellable graphite had a positive influence over the thermal stability and flame retardant performance of the composites [12].”, it has now been revised and supported with latest literatures as follows.

There are scholars using different fillers to change the properties of the composites in recent years [13-18].

[13]. Lee, Heow Pueh, et al. "An investigation of the sound absorption properties of flax/epoxy composites compared with glass/epoxy composites." Journal of Natural Fibers 14.1 (2017): 71-77.

[14]. Khaleel, Mustafa, Ugur Soykan, and Sedat Çetin. "Influences of turkey feather fiber loading on significant characteristics of rigid polyurethane foam: Thermal degradation, heat insulation, acoustic performance, air permeability and cellular structure." Construction and Building Materials 308 (2021): 125014.

[15]. Mohammadi, Behzad, et al. "Mechanical and sound absorption properties of open-cell polyurethane foams modified with rock wool fiber." Journal of Building Engineering (2021): 103872.

[16]. Li, Ting-Ting, et al. "Preparation and characteristics of flexible polyurethane foam filled with expanded vermiculite powder and concave-convex structural panel." Journal of Materials Research and Technology 12 (2021): 1288-1302.

[17]. Imran, Mohammed, et al. "Mechanical property enhancement of flexible polyurethane foam using alumina particles." Materials Today: Proceedings 45 (2021): 4040-4044.

[18]. Choe, Hyeon, Jae Heon Lee, and Jung Hyeun Kim. "Polyurethane composite foams including CaCO3 fillers for enhanced sound absorption and compression properties." Composites Science and Technology 194 (2020): 108153.

  1. Discussion: The discussion is still very generic. This topic required wider discussion with the up-to-date literature, compare obtained results with other research works, add new literature.

Answer: Thank you for your comment. More latest literatures have now been appended to the sections 3.1, 3.2, 3.4, and 3.6 for comparison and support as follows. Additionally, section 3.5 has now been incorporated with further explanation with a supportive literature. All of the newly appended literatures have now been supplemented to the reference.

In section 3.1:

[25]. Baek, Seung Hwan, and Jung Hyeun Kim. "Polyurethane composite foams including silicone-acrylic particles for enhanced sound absorption via increased damping and frictions of sound waves." Composites Science and Technology 198 (2020): 108325.

[26]. Park, Sung Jun, et al. "Natural cork agglomerate enabled mechanically robust rigid polyurethane foams with outstanding viscoelastic damping properties." Polymer 217 (2021): 123437.

In section 3.2:

[16]. Li, Ting-Ting, et al. "Preparation and characteristics of flexible polyurethane foam filled with expanded vermiculite powder and concave-convex structural panel." Journal of Materials Research and Technology 12 (2021): 1288-1302.

[27]. Wang, Wei, et al. "Rigid polyurethane foams based on dextrin and glycerol." Industrial Crops and Products 177 (2022): 114479.

[28].Leszczyńska, Milena, et al. "Cooperative effect of rapeseed oil-based polyol and egg shells on the structure and properties of rigid polyurethane foams." Polymer Testing 90 (2020): 106696.

In section 3.4:

[16]. Li, Ting-Ting, et al. "Preparation and characteristics of flexible polyurethane foam filled with expanded vermiculite powder and concave-convex structural panel." Journal of Materials Research and Technology 12 (2021): 1288-1302.

[17]. Imran, Mohammed, et al. "Mechanical property enhancement of flexible polyurethane foam using alumina particles." Materials Today: Proceedings 45 (2021): 4040-4044.

[25]. Baek, Seung Hwan, and Jung Hyeun Kim. "Polyurethane composite foams including silicone-acrylic particles for enhanced sound absorption via increased damping and frictions of sound waves." Composites Science and Technology 198 (2020): 108325.

In section 3.5:

On the other hand, the incorporation of filler is also pertinent to the viscoelasticity of PU foam [26], so the viscoelasticity of cell walls is dependent on the presence of HCM. Furthermore, a rise in the content of filler may lead to agglomeration that interferes the impact resistance of the composites as previously described.

[26]. Park, Sung Jun, et al. "Natural cork agglomerate enabled mechanically robust rigid polyurethane foams with outstanding viscoelastic damping properties." Polymer 217 (2021): 123437.

In section 3.6:

[13]. Lee, Heow Pueh, et al. "An investigation of the sound absorption properties of flax/epoxy composites compared with glass/epoxy composites." Journal of Natural Fibers 14.1 (2017): 71-77.

[14]. Khaleel, Mustafa, Ugur Soykan, and Sedat Çetin. "Influences of turkey feather fiber loading on significant characteristics of rigid polyurethane foam: Thermal degradation, heat insulation, acoustic performance, air permeability and cellular structure." Construction and Building Materials 308 (2021): 125014.

[15]. Mohammadi, Behzad, et al. "Mechanical and sound absorption properties of open-cell polyurethane foams modified with rock wool fiber." Journal of Building Engineering (2021): 103872.

[18]. Choe, Hyeon, Jae Heon Lee, and Jung Hyeun Kim. "Polyurethane composite foams including CaCO3 fillers for enhanced sound absorption and compression properties." Composites Science and Technology 194 (2020): 108153.

[25]. Baek, Seung Hwan, and Jung Hyeun Kim. "Polyurethane composite foams including silicone-acrylic particles for enhanced sound absorption via increased damping and frictions of sound waves." Composites Science and Technology 198 (2020): 108325.

[31]. Sukhawipat, Nathapong, et al. "Effects of water hyacinth fiber size on sound absorption properties of advanced recycled palm oil-based polyurethane foam composite." Materials Today: Proceedings (2021).

Round 3

Reviewer 1 Report

The revised manuscript - entitled: “Functional Hollow Ceramic Microsphere/Flexible Polyurethane Foam Composites with a Cell Structure: Mechanical Property and Sound Absorptivity” were improved by the authors by the adding some new literature. However still is missing some detailed comments about the provided research and comparison with other research work. The authors should firstly present the description of their own research and next compare it with other works. Are the same results / main findings or different? What is the reason of differences?

Author Response

R3 polymers-1573690

表單的頂端

Journal

Polymers (ISSN 2073-4360)

Manuscript ID

polymers-1573690表單的底部

Revised3

Comments and Suggestions for Authors

The revised manuscript - entitled: “Functional Hollow Ceramic Microsphere/Flexible Polyurethane Foam Composites with a Cell Structure: Mechanical Property and Sound Absorptivity” were improved by the authors by the adding some new literature. However still is missing some detailed comments about the provided research and comparison with other research work. The authors should firstly present the description of their own research and next compare it with other works. Are the same results / main findings or different? What is the reason of differences?

Thank you very much for your insightful advices and recommendations. We would like to express our thankfulness to the reviewer for your valuable suggestions to improve the quality of the manuscript.

Answer: We have highlighted these parts in section 3.5 (Line 241-Line 244) . Therefore, there was no additional discussion.

Answer: In order to make it clear, we have added discussions in section 3.4 and 3.6.

Answer: We have added literatures (Baek, Seung et al.) to explain the mechanism of STF in sextion 3.4 (line 195).

On the other hand, The introduction of inorganic filler was found to be a critical factor improving the mechanical property (in line 200). These conclusions was confirmed with the Mohammed Imran work. Notably, scholars have yet to use HCM as form fillers to improve the material property (Detailed information can be found in introduction). As a result, the suggested relationship between HCM and mechanical performance could be one directive principle for designing desired durable form materials.

Answer: We have added discussions in section 3.6 (line 274) . Modify as follows:

When the filler content increased, this study simialrs to the previous work (Baek, Seung Hwan. [25]) in sound absorption property. The sound absorption frequency increases in medium high frequency with further increments of filler content. In addition, the relationship between sound absorption coefficient and form structure were investigated in line 277. The results were compared with the privious work (Sung et al. [25], in line 283).

,
